human-computer interaction/biomechanics/ bioengineering

human balancing, reaction delay, delayed feedback, stabilizability, robustness

**Author for correspondence:**
Tamas Insperger
e-mail: insperger@mm.bme.hu

# Virtual stick balancing: sensorimotor uncertainties related to angular displacement and velocity

Balazs A. Kovacs[1], John Milton[2] and Tamas Insperger[1]

[1]Department of Applied Mechanics, Budapest University of Technology and Economics and MTA-BME Lendület Human Balancing Research Group, Budapest, Hungary
[2]W. M. Keck Science Department, The Claremont Colleges, Claremont, CA 91711, USA

 BAK, 0000-0003-2942-730X; JM, 0000-0001-5037-2548;
TI, 0000-0001-7518-9774

Sensory uncertainties and imperfections in motor control play important roles in neural control and Bayesian approaches to neural encoding. However, it is difficult to estimate these uncertainties experimentally. Here, we show that magnitude of the uncertainties during the generation of motor control force can be measured for a virtual stick balancing task by varying the feedback delay, $\tau$. It is shown that the shortest stick length that human subjects are able to balance is proportional to $\tau^2$. The proportionality constant can be related to a combined effect of the sensory uncertainties and the error in the realization of the control force, based on a delayed proportional-derivative (PD) feedback model of the balancing task. The neural reaction delay of the human subjects was measured by standard reaction time tests and by visual blank-out tests. Experimental observations provide an estimate for the upper boundary of the average sensorimotor uncertainty associated either with angular position or with angular velocity. Comparison of balancing trials with 27 human subjects to the delayed PD model suggests that the average uncertainty in the control force associated purely with the angular position is at most 14% while that associated purely with the angular velocity is at most 40%. In the general case when both uncertainties are present, the calculations suggest that the allowed uncertainty in angular velocity will always be greater than that in angular position.

## 1. Introduction

Sensory feedback is necessary for the skilled performance of novel voluntary motor tasks [1–3]. With practice extending over weeks to years, the nervous system gradually develops an internal

model which predicts the sensory consequences of the voluntary movements [4–7]. As skill increases there is a decrease in the variability in the outcome of repetitions of the motor task [8]. However, during daily living, humans perform many voluntary tasks which have not been well learned or even practised once. Even though the nervous system may partly rely on motor programs previously learned for the performance of similar motor tasks, there can still be considerable uncertainty associated with the application of these motor programs to the new task. The control problems are particularly difficult on the first day of practice since there has been no opportunity for sleep-dependent consolidation and refinement of motor control [9,10]. Understanding the control of these unpractised motor tasks has important implications ranging from the design of teaching strategies to the design of automobiles to safely perform a sudden manoeuvre during driving or in airplanes to facilitate an emergency evacuation.

Tasks controlled by feedback are constrained by the reaction time delay and sensorimotor uncertainties in the control process, such as sensory uncertainties and error in motor control realization (motor noise). Although these uncertainties cannot be controlled, they nonetheless place constraints on the nature of the control mechanisms that work best in unpredictable environments. Recently emphasized examples arise in the control of autonomous vehicles [11,12] and in the role played by sensory dead zones in balance control [6,13]. In addition sensorimotor uncertainties play a central role in Bayesian approaches to neural encoding [14,15] and, if large enough, undermine the effectiveness of internal models to predict the sensory consequences of movements. These uncertainties can arise in many ways including the effects of temporal and spatial quantization of the sensory inputs [16,17], static uncertainties related to the system and control parameters [18–20], and noise [21–26]. Neuroimaging studies emphasize that the disparity between novice and expert motor performance lies at the level of the organization of the neural networks that are involved in motor planning [27,28]. Motor planning neural networks in novices are more diffusely and extensively activated compared to those in experts. Moreover, novices activate certain brain regions, such as the posterior cingulate, suggesting that they have difficulty filtering out irrelevant information. In terms of motor control theory, these observations suggest that uncertainties in generating the control force are important features of the motor control process.

Whereas the reaction time delay, $\tau$, can be readily estimated using reaction time tests [5,6,29–31] it has been difficult to obtain estimates of uncertainties in the control force generation. An exception arises in the control of unstable states, in particular, stick balancing on the fingertip [20]. In particular, these studies have drawn attention to the importance of determining the shortest stick length, $L_{\mathrm{crit}}$, that can be balanced for a given $\tau$. Analytical and numerical simulations show that $L_{\mathrm{crit}}$ increases as $\tau^2$ [20,32]. Sensorimotor uncertainties impair control performance, hence increases $L_{\mathrm{crit}}$ further. Virtual stick balancing tasks [21,26,33–35] that involve the interplay between a human and a computer provide an environment in which $\tau$ can be readily manipulated. Thus it becomes possible, at least in principle, to directly measure the dependence of $L_{\mathrm{crit}}$ on $\tau$ and use this dependence to obtain insight into the magnitude of the sensory uncertainties related to the angular displacement and velocity.

In the engineering literature, static uncertainties play an important role in the design of effective control strategies in uncertain environments [36,37]. A good measure for the robustness of the controller against static uncertainties is the stability radius, which is a kind of 'distance from instability' and hence is related to the size of the stable domain in the space of the control and system parameters. The concept of stability radius has been already adopted in human balance models [18,19] and it is also used in this paper for robust stability analysis.

The paper is organized as follows. First, the concepts of $L_{\mathrm{crit}}$ and sensorimotor uncertainties in relation to the control of an inverted pendulum by time-delayed proportional-derivative (PD) feedback are discussed. In §3, we describe the construction of a virtual stick balancing task and in §4, we describe the results obtained for 27 novices during one day of practice. It is observed that the allowed sensorimotor uncertainties related to the angular velocity are larger than those related to the angular displacement.

## 2. Background

Controlling an inverted pendulum in the presence of feedback delay is a benchmark problem in control systems theory and is often used to model human balancing [38,39]. It is known that upper equilibrium of the pendulum can be stabilized by a PD controller if the feedback delay is less than a critical delay $\tau_{\mathrm{crit}} = T/(\pi\sqrt{2})$, where $T$ is the oscillation period of the structure hung at its downward position [40]. The governing equation for a pendulum with a uniformly distributed mass balanced on a cart is

$$\ddot{\varphi}(t) - a\varphi(t) = -p\varphi(t - \tau) - d\dot{\varphi}(t - \tau), \qquad (2.1)$$

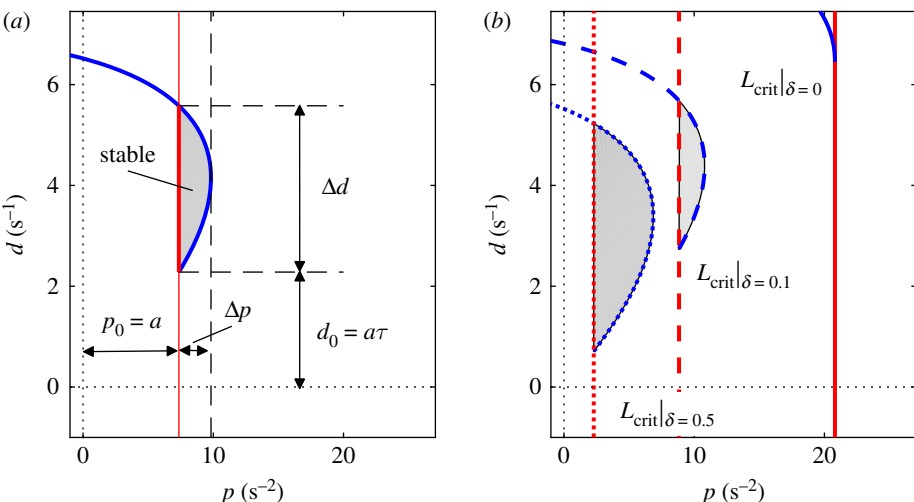

**Figure 1.** (a) Stability chart for delayed PD feedback for $\tau = 0.3$ s and $L = 2$ m. Red and blue lines denote real root boundary and complex root boundary, respectively. Stabilizing control gains are indicated by grey shading. Robustness of the controller is related to the size of the stable region characterized by $\Delta p$ and $\Delta d$. (b) Stability boundaries for the critical length associated with stability radii $\delta_p = 0$ (dotted), $\delta_p = 0.1$ (dashed) and $\delta_p = 0.5$ (solid).

where $\varphi$ is the angular position measured from the $\varphi = 0$ vertical upright position, $a = 3g/(2L)$ is a system parameter inversely proportional to the length $L$, $g$ is the acceleration due to gravity, $p$ and $d$ are, respectively, the proportional and derivative control gains and $\tau$ is the reaction delay. The oscillation period about the lower equilibrium is $T = 2\pi\sqrt{2L/(3g)}$, hence, $\tau_{\text{crit}} = \sqrt{4L/(3g)}$.

## 2.1. Stability

The stability chart of (2.1) can be determined by the D-subdivison method [41] after substituting $\lambda = i\omega$ into the characteristic equation

$$\lambda^2 - a + pe^{-\lambda\tau} + d\lambda e^{-\lambda\tau} = 0. \tag{2.2}$$

The so-called D-curves are the transition curves in the parameter plane $(p, d)$ where characteristic exponents exist with zero real part. Figure 1 shows the D-curves and the stable parameter region for (2.1). Two types of D-curves can be distinguished. The line $p = a$ shown by red line in figure 1 is associated with a real characteristic root when $\omega = 0$. This type of D-curve is often referred to as real root boundary (RRB) in the literature [42]. The parametric curve

$$p(\omega) = (a + \omega^2)\cos(\omega\tau) \tag{2.3}$$

and

$$d(\omega) = \frac{a + \omega^2}{\omega}\sin(\omega\tau) \tag{2.4}$$

shown by blue line in figure 1 is associated with a pair of pure imaginary characteristic roots $\lambda = \pm i\omega$. This type of D-curve is called complex root boundary (CRB). The parametric curve (2.3)–(2.4) is initiated from the parameter point $(p_0, d_0)$, where

$$p_0 = \lim_{\omega \to 0} p(\omega) = a \tag{2.5}$$

and

$$d_0 = \lim_{\omega \to 0} d(\omega) = a\tau. \tag{2.6}$$

This point provides a useful reference point in the plane $(p, d)$.

## 2.2. Theoretical critical length

It is known that feedback delay limits the stabilizability of control systems. In the stick balancing problem, this limitation can be represented by the critical length $L_{\text{crit}}$. For a given feedback delay $\tau$,

the critical length is $L_{\mathrm{crit}} = (3/4)g\tau^2$ (e.g. [32,40]). If the pendulum is shorter than $L_{\mathrm{crit}}$ then it cannot be stabilized about its upside equilibrium position using a PD control with feedback delay $\tau$. This relation explains why it is more difficult to balance a short pendulum than a long one. The panels in figure 1 demonstrate the concept of the critical length. If the feedback delay $\tau$ is fixed and the length $L$ of the pendulum is decreased, then the size of the stable regions decreases. When $L = L_{\mathrm{crit}}$, then the stable region disappears at the parameter point $(p_0, d_0)$ as shown by the solid lines in figure 1b.

## 2.3. Critical length in the presence of static sensorimotor uncertainties

When $L > L_{\mathrm{crit}}$, then the system is stable if the control gains are selected from the shaded area in figure 1. The larger the stable region, the larger the robustness against the error in selecting $p$ and $d$. The size of the stable region along the $p$ and the $d$ directions can be characterized by the stability radii

$$\delta_{\mathrm{p}} = \frac{(1/2)\Delta p}{p_0 + (1/2)\Delta p} \quad \text{and} \quad \delta_{\mathrm{d}} = \frac{(1/2)\Delta d}{d_0 + (1/2)\Delta d}, \tag{2.7}$$

where $\Delta p$ and $\Delta d$ are the width and the height of the stable region. If the control gains are tuned to the middle of the stable region, i.e. $p = p_0 + (1/2)\Delta p$ and $d = d_0 + (1/2)\Delta d$, then larger than $\delta_{\mathrm{p}}$ relative error in the tuning of $p$ or larger than $\delta_{\mathrm{d}}$ relative error in the tuning of $d$ destabilizes the system. Parameters $\Delta p$ and $\Delta d$ at the same time show the robustness of the system against static sensorimotor uncertainties [18,19,36,37]. Figure 1b shows the robust stability boundaries associated with $\delta_{\mathrm{p}} = 0$, $\delta_{\mathrm{p}} = 0.1$ and $\delta_{\mathrm{p}} = 0.5$ by solid, dashed and dotted lines, respectively. The corresponding critical lengths are, respectively, $L_{\mathrm{crit}}|_{\delta=0} = 0.707\,\mathrm{m}$, $L_{\mathrm{crit}}|_{\delta=0.1} = 1.665\,\mathrm{m}$, $L_{\mathrm{crit}}|_{\delta=0.5} = 6.432\,\mathrm{m}$. This means that in the case of 10% static uncertainty in the gain $p$, stable control for a stick of length shorter than $1.665\,\mathrm{m}$ cannot be guaranteed by delayed PD feedback. The case of $L_{\mathrm{crit}}|_{\delta=0}$ corresponds to the case when the stable region bounded by the D-curves (2.3)–(2.4) just disappears.

In order to analyse the effect of $\Delta p$ and $\Delta d$ separately, the control force is decomposed into components as $Q(t) = Q_{\mathrm{p}}(t) + Q_{\mathrm{d}}(t)$ where $Q_{\mathrm{p}}(t) = p\varphi(t - \tau)$ and $Q_{\mathrm{d}}(t) = d\dot{\varphi}(t - \tau)$ are associated with proportional and derivative feedback, respectively. In the case of static error $\Delta\varphi = \varepsilon_{\mathrm{p}}\varphi$ in the perception of $\varphi$, the corresponding control force becomes

$$\tilde{Q}_{\mathrm{p}}(t) = p\left(\varphi(t - \tau) + \Delta\varphi(t - \tau)\right) = (1 + \varepsilon_{\mathrm{p}})\,p\,\varphi(t - \tau). \tag{2.8}$$

Similarly, static error $\Delta\dot{\varphi} = \varepsilon_{\mathrm{d}}\dot{\varphi}$ in the perception of $\dot{\varphi}$ alters the control force term $Q_{\mathrm{d}}$ as

$$\tilde{Q}_{\mathrm{d}}(t) = d\left(\dot{\varphi}(t - \tau) + \Delta\dot{\varphi}(t - \tau)\right) = (1 + \varepsilon_{\mathrm{d}})\,d\,\dot{\varphi}(t - \tau). \tag{2.9}$$

Static error $\Delta Q = \varepsilon_{\mathrm{Q}}\tilde{Q}$ in the realization of the control force can also be represented as a multiplicative error in the actual control force: $\hat{Q}(t) = (1 + \varepsilon_{\mathrm{Q}})\tilde{Q}(t)$. The corresponding terms of the actual control force then read

$$\hat{Q}_{\mathrm{p}}(t) = (1 + \varepsilon_{\mathrm{Q}})\,\tilde{Q}_{\mathrm{p}}(t) = (1 + \varepsilon_{\mathrm{Q,p}})\,p\,\varphi(t - \tau) \tag{2.10}$$

and

$$\hat{Q}_{\mathrm{d}}(t) = (1 + \varepsilon_{\mathrm{Q}})\,\tilde{Q}_{\mathrm{d}}(t) = (1 + \varepsilon_{\mathrm{Q,d}})\,d\,\dot{\varphi}(t - \tau), \tag{2.11}$$

where $\varepsilon_{\mathrm{Q,p}} = \varepsilon_{\mathrm{p}} + \varepsilon_{\mathrm{Q}} + \varepsilon_{\mathrm{p}}\varepsilon_{\mathrm{Q}}$ and $\varepsilon_{\mathrm{Q,d}} = \varepsilon_{\mathrm{d}} + \varepsilon_{\mathrm{Q}} + \varepsilon_{\mathrm{d}}\varepsilon_{\mathrm{Q}}$. Here $\varepsilon_{\mathrm{p}}$, $\varepsilon_{\mathrm{d}}$ and $\varepsilon_{\mathrm{Q}}$ are the uncertainty ratios associated with the perception of the position and the velocity and with the realization of the control force, while $\varepsilon_{\mathrm{Q,p}}$ and $\varepsilon_{\mathrm{Q,d}}$ describe their combined effect. If the combined uncertainty ratio $\varepsilon_{\mathrm{Q,p}}$ is larger then the corresponding stability radius $\delta_{\mathrm{p}}$, then the closed-loop system becomes unstable. Similarly, if $\varepsilon_{\mathrm{Q,d}} > \delta_{\mathrm{d}}$, then the closed-loop system is again unstable. This way, the stability radii $\delta_{\mathrm{p}}$ and $\delta_{\mathrm{d}}$ are directly related to the combined static errors $\varepsilon_{\mathrm{Q,p}}$ and $\varepsilon_{\mathrm{Q,d}}$, respectively. Thus, $\Delta p$ and $\Delta d$ reflect the robustness against the static uncertainty in the perception of the angular position $\varphi$ and the angular velocity $\dot{\varphi}$, respectively, in combination with the robustness against the realization of the control force $Q$.

The critical length for a given stability radius $\delta_{\mathrm{p}}$ or $\delta_{\mathrm{d}}$ is the length $L$ where the width or the height of the stable region is just equal to $p_0(1 + 2\delta_{\mathrm{p}})$ or $d_0(1 + 2\delta_{\mathrm{d}})$, respectively. The critical lengths associated with different stability radii $\delta_{\mathrm{p}}$ and $\delta_{\mathrm{d}}$ as function of reaction time delay are shown in figure 2a,b. The curves associated with $\delta_{\mathrm{p}} = 0$ and $\delta_{\mathrm{d}} = 0$ are given by $L_{\mathrm{crit}}|_{\delta=0} = a_0\tau^2$ where $a_0 = (3/4)g$ and the condition $\delta = 0$ refers to the theoretical case without uncertainties (see §2.2). It can be shown by

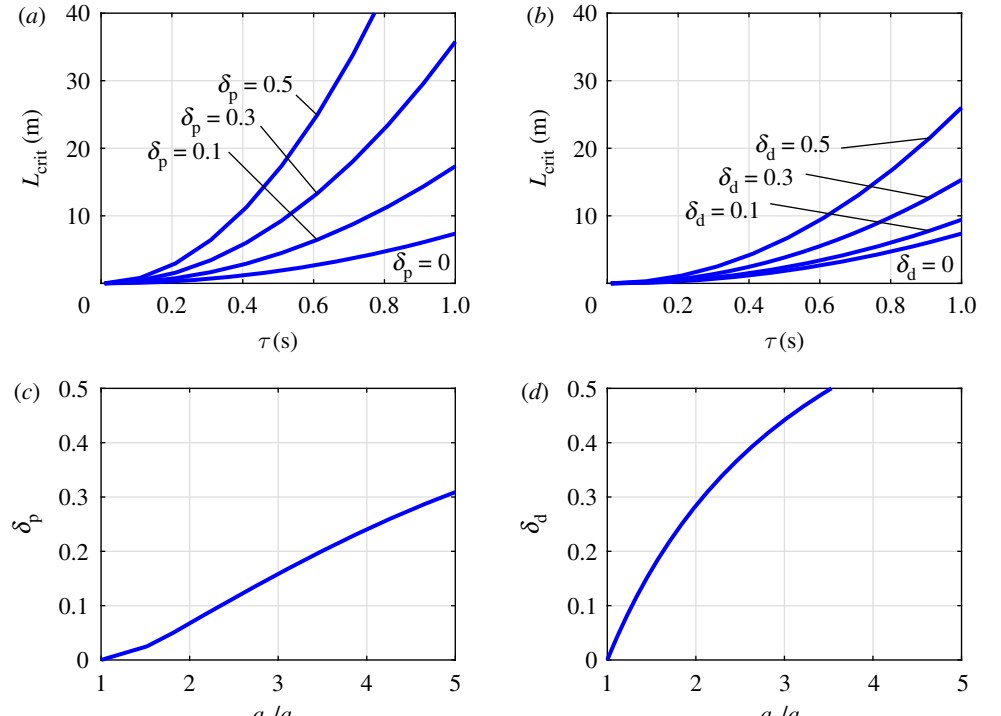

**Figure 2.** Critical length as function of the reaction delay for different stability radii $\delta_p$ (a) and $\delta_d$ (b). The stability radii $\delta_p$ (c) and $\delta_d$ (d) as function of the ratio $a_1/a_0$.

analytical manipulation of (2.3) and (2.4) that the critical length associated with $\delta_p > 0$ or $\delta_d > 0$ can be written in the form $L_{\mathrm{crit}}|_{\delta>0} = a_1\tau^2$, where coefficient $a_1$ describes the contribution of the static error in $p$ and $d$. As (2.10) and (2.11) show, the permissible error in $p$ and $d$ can be associated with the combined uncertainties $\varepsilon_{Q,p}$ and $\varepsilon_{Q,d}$. Thus, if $\varepsilon_{Q,p} > \delta_p$ or $\varepsilon_{Q,d} > \delta_d$ then the control system is unstable. In this sense, coefficient $a_1$ describes the contribution of the combined uncertainties $\varepsilon_{Q,p}$ and $\varepsilon_{Q,d}$ to the increase of the critical length. The stability radii $\delta_p$ and $\delta_d$ as function of the ratio $a_1/a_0$ are shown in figure 2c,d. Note that $\delta_d$ is larger than $\delta_p$ for the same ratio $a_1/a_0$, which indicates that larger relative error is allowed in $d$ than in $p$. This property is implied by the shape of the stable region of the delayed PD feedback model.

Taken together these observations demonstrate that the static sensorimotor uncertainties in the control process can be estimated by measuring $L_{\mathrm{crit}}$ as a function of $\tau$. Specifically, the coefficient $a_1$ is determined by fitting a second-order curve onto the measured critical lengths. Then the corresponding stability radii $\delta_p$ or $\delta_d$ can be determined according to figure 2c,d, respectively. Here, we use this approach to estimate the combined sensorimotor uncertainty for a virtual stick balancing task for subjects during the first day of practice.

# 3. Methods

## 3.1. Mechanical model

The mechanical model of an inverted pendulum is used, where the suspension point is manipulated by the human subject. The corresponding equation of motion reads

$$\frac{1}{3}L^2\ddot{\varphi}(t) - \frac{1}{2}Lg\sin\varphi = -\frac{1}{2}La_S(t)\cos\varphi, \tag{3.1}$$

where $a_S$ is the acceleration of the suspension point. This equation is used for the real-time simulations during the virtual balancing tests.

Stability and stabilizability properties are analysed by assuming a delayed PD feedback in the form

$$a_S(t) = k_p\varphi(t-\tau) + k_d\dot{\varphi}(t-\tau), \tag{3.2}$$

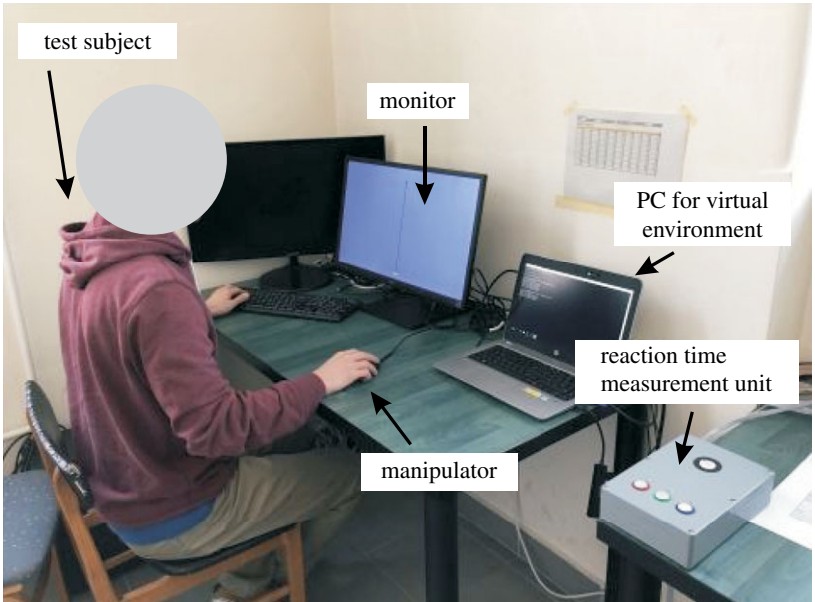

**Figure 3.** The measurement set-up of the virtual stick balancing experiments.

where $k_p$ and $k_d$ are the actual control gains for the acceleration $a_S$. After linearization, equation (3.1) can be written in the form of equation (2.1) with $p = 3k_p/(2L)$ and $d = 3k_d/(2L)$.

## 3.2. Participants

Twenty-seven subjects were recruited from the local student and faculty population (age between 19 and 42 years, 23 in average, six females, 21 males). All subjects were free of any neurological or musculoskeletal impairment that could affect virtual stick balancing. The research was carried out following the principles of the Declaration of Helsinki and was approved by the Hungarian National Science and Research Ethics Committee. All participants provided informed consent for all research testing and were given the opportunity to withdraw from the study at any time.

## 3.3. Reaction delay measurement

Two classic forms of reaction delay test were used [29–31]. In the first task (referred to as 'SINGLE'), the subject kept their index finger on a button and pressed it as quickly as possible in response to a single light flash. In the second task (referred to as 'INDIVIDUAL'), the subject kept their index and middle fingers on two nearby buttons and pressed the button which corresponded to a single light flash as quickly as possible. Ten trials were performed for both tests such that the time increments between the succeeding flashes were randomized. More details on the reaction time test equipment used in the test is given in [43].

## 3.4. Virtual stick balancing environment

The virtual stick balancing environment is shown in figure 3. The software for the virtual stick balancing task was developed in a JAVA environment. The governing equation (3.1) of the inverted pendulum was solved using fourth-order Runge–Kutta method with adaptive time step such that the simulation was running in real time. The interface for the virtual environment was a conventional optical computer mouse. For the actuation of the virtual environment, the input signal is the acceleration of the computer mouse moved by the subject's hand. The acceleration was determined via numerical derivation of the pixel-based position of the cursor [44]. Due to the finite number of pixels, this results in a noisy acceleration signal. Therefore, a simple re-sampling filter was used, and the input acceleration was computed as

$$a_S(t_i) = K \frac{x(t_i) - 2x(t_{i-k}) + x(t_{i-2k})}{\Delta t^2}, \tag{3.3}$$

where $K = 8.85 \times 10^{-5}$ m/pixel is a gain factor scaling the screen size to the manipulation length of the computer mouse [44], $x(t_i)$ is the mouse position measured in pixels at the time instant $t_i = i\Delta t$ and

$k \geq 1$ is an integer filtering parameter. The sampling period was set to $\Delta t = 16.67$ ms, which corresponds to the screen refresh rate 60 Hz. The acceleration $a_S(t_i)$ and the stick's position $\varphi(t_i)$ and velocity $\dot{\varphi}(t_i)$ were computed within a single sampling period.

A HP Probook 430 was connected to a 24-inch LG 24BK550Y-B monitor via VGA cable. The vertical dimension was scaled such that all sticks appeared to be 23 cm long on the computer screen. The displacements in the horizontal direction were not scaled in order to match the real deviation of the underling dynamical model. This way, the displacement between the top and the bottom of the stick appeared in its real size on the computer screen. Subjects were instructed to concentrate on the top of the stick during the stick balancing trials.

## 3.5. Machine delay and delay increments

The response time of the computer screen, signal filtering and the screen refresh rate introduce a machine delay $\tau_M$ which is equal to the time between when an input is produced by a computer mouse and its appearance on the screen. Different computer-screen configurations have been investigated to make the machine delay as small as possible [44]. The screen response time was measured using a light sensor system, which detects black and white transition time of the screen synchronized to the mouse input. The response time between a mouse input and its full representation on the screen was measured to be $\tau_{M,response} = 64$ ms. Signal filtering for the acceleration (3.3) introduces a large artificial delay $\tau_{M,filter} = k\Delta t$. The optimal value for the filtering parameter was experimentally found to be $k = 3$, which gives the additional delay $\tau_{M,filter} \cong 50$ ms. Finally, the time step $\Delta t$ for the simulation was adjusted to the screen refresh rate, which was set to 60 Hz. This sampling effect introduces a delay which varies linearly between 0 and $\Delta t$ with an average of $\tau_{M,sampling} = \Delta t / 2 = 8.3$ ms. Thus the total machine delay is

$$\tau_M = \tau_{M,response} + \tau_{M,sampling} + \tau_{M,filter} \cong 122 \text{ ms}. \tag{3.4}$$

The overall reaction delay for virtual stick balancing is

$$\tau = \tau_N + \tau_M + \tau_{added}, \tag{3.5}$$

where $\tau_N$ is the neural reaction delay of human subjects and $\tau_{added}$ is the artificially added delay. During the virtual balancing tests, $\tau_{added}$ was increased in steps of $\Delta\tau = 50$ ms.

## 3.6. Blank-out tests

Reaction delay during virtual stick balancing is measured from the response to a visual blank-out [6]. The stick disappears from the screen for a period of length 500 ms at a random time instant between 5 and 10 s after the start of the trial. The first corrective motion after the end of the blank-out indicates the length of the reaction time delay of the subject. In order to get an objective estimate for the delay for all the blank-out trials, a sweeping window technique was constructed using two time windows $W_b = [t_i - \Delta w, t_i]$ and $W_a = [t_i, t_i + \Delta w]$ before and after the time instant $t_i$ with length $\Delta w = 300$ ms. The indication function

$$\Pi(t_i) = \frac{(m_b(t_i) - m_a(t_i))^2}{m_b(t_i) + m_a(t_i)} \tag{3.6}$$

was defined to indicate the change in the corrective acceleration, where

$$m_b(t_i) = \frac{1}{N} \sum_{i=-N}^{1} |a(t_i)| \quad \text{and} \quad m_a(t_i) = \frac{1}{N} \sum_{i=1}^{N} |a(t_i)| \tag{3.7}$$

are the absolute mean values of the measured accelerations over the windows $W_b$ and $W_a$ with $N = \Delta w / \Delta t$. The peak in $\Pi$ indicates the time instant where the largest change in the corrective acceleration takes place.

A sample measurement data of a blank-out test can be seen in figure 4. The top panel presents the measured corrective acceleration. The time is scaled such that the end of the 500 ms blank-out is at $t = 0$ s. The blank-out (off) period is indicated by grey shading. Red curves show the individual trials, while their average is shown by blue lines. It can clearly be seen that significant change in the corrective acceleration takes place at about 270 ms after the end of the blank-out. Since the acceleration of the computer mouse is readily measured, this time corresponds to the subjects' reaction delay $\tau_N$. It should be noted that the corrective accelerations before the blank-out and during the blank-out are in the same range while after the blank-out significantly larger correction can be observed. This shows that after the blank-out a huge corrective action is necessary to keep the stick balanced (see also fig. 2 in [6]).

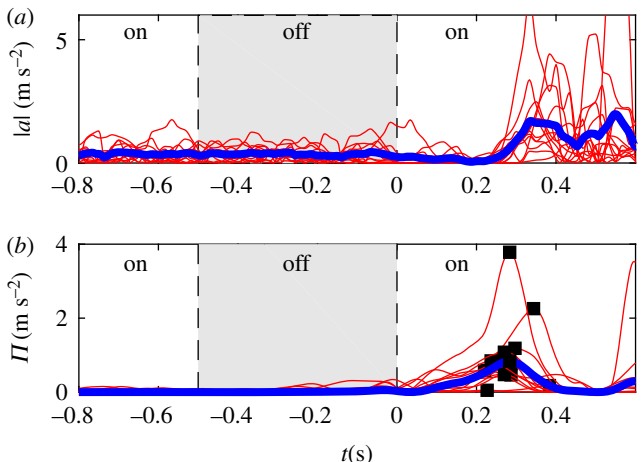

**Figure 4.** Sample of a blank-out test: corrective acceleration (*a*) and the indicator function $\Pi$ (*b*). Corrective action takes place when $\Pi$ reaches a maximum.

## 3.7. Balancing protocol

Each subject was tested as follows:

1. Reaction time test SINGLE: 10 trials.
2. Reaction time test INDIVIDUAL: 10 trials.
3. Practice with a 5 m long virtual stick without extra added delay: 10 min.
4. The critical length was determined by a halving method. First, the subject performed five trials without added delay with the initial length $L_0$ that the subject was able to balance without any difficulty during the 10 min practice. The balancing of a stick of length $L_0$ was deemed 'Successful' if a balance time of 20 s was achieved such that the stick's angle never exceeded $\pm 20°$ for at least one out of the five trials [6]. If the balancing was successful/unsuccessful, then the $L$ was decreased/increased by $\Delta L_1 = 1$ m and the subjects did five trials. If successful/unsuccessful, the stick length was decreased/increased by $\Delta L_2 = 0.5$ m and the subject did five trials, and so on. After five steps, the last change in the stick length was $\Delta L_5 = 0.0625$ m. The critical length $L_{crit}(0)$ was determined as the shortest stick length with successful balancing.
5. The time delay was increased by adding $\tau_{added} = \Delta\tau = 50$ ms to the feedback and Step 4 was repeated with initial length $L_0 = L_{crit}(0) + 1$ m. In this way, $L_{crit}(50)$ was determined for an added delay of 50 ms.
6. Step 5 was repeated for added delays of $\tau_{added} = 100, 150, 200, 250$ and 300 ms to obtain, respectively, $L_{crit}(100), L_{crit}(150), L_{crit}(200), L_{crit}(250)$ and $L_{crit}(300)$.
7. Finally, a blank-out test [6] was made using a stick length which was well balanced by the subject. This was typically $L_{crit}(0) + 1$ m. The blank-out test provides an estimate of the subject's neural reaction delay $\tau_N$.

It typically took 60 min for a subject to complete this balancing protocol. All the subjects were able to do the balancing tasks for all added delay from 0 to 300 ms successfully. No subject complained of fatigue.

## 4. Results

The data of the measurements are available within the Dryad Digital Repository: https://doi.org/10.5061/dryad.41ns1rn9m [45].

## 4.1. Reaction delays

All the test subjects were able to perform all the three types of reaction time tests (SINGLE, INDIVIDUAL, BLANK-OUT) successfully. The results of the different tests are summarized in figure 5. The shortest mean reaction delay (227 ms in average) was obtained in the SINGLE reaction time test, while the INDIVIDUAL

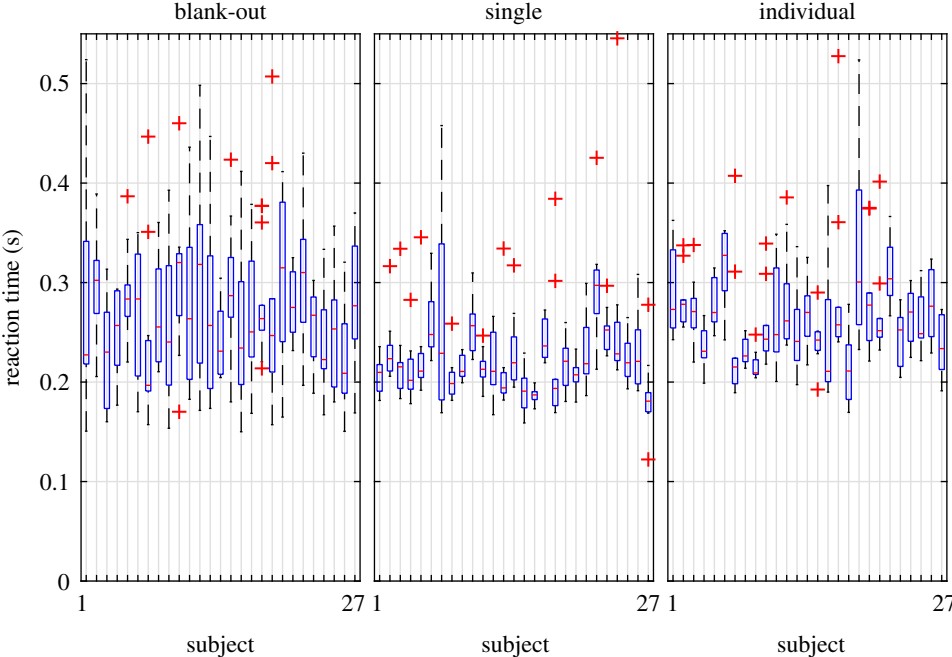

**Figure 5.** Boxplots of the different reaction time tests for the 27 subjects.

tests resulted in slightly larger mean reaction times (258 ms in average). The mean reaction times in the BLANK-OUT test (270 ms in average) correspond to that of the INDIVIDUAL tests (24 out of the 27 subjects passed the modified $t$-test at 98% significance level). A repeated measure ANOVA test was also performed in order to check the relation between the results of different reaction time measurements. The test showed a statistically significant relation between the means of the reaction times at 94% significance level. This observation supports the assumption that both the INDIVIDUAL test and the virtual stick balancing involve a decision making, while in the SINGLE reaction tests, subjects just responded to a signal in a simple way.

The variance of the reaction times in the BLANK-OUT test is significantly larger than for the other two tests. This is due to the complex concept of the BLANK-OUT test and due to the equivocal evaluation of the change in the corrective acceleration. For instance, the stick may happen to be close to the vertical position by the end of the blank-out period just by pure chance and the subjects do not have to react sharply. On the other hand, subjects sometimes react before the return of the visual feedback although they were instructed not to do so. This action might be driven by some internal predictive processes, i.e. the subjects think/estimate that they must do some action otherwise the stick will fall even though they do not know the actual position of the stick. These effects cause an uncertainty in the evaluation of the reaction time by the indicator function $\Pi$, which is reflected in the larger variance of the BLANK-OUT tests. Nevertheless figure 5 shows that the deviation of the measured reaction time delays increases with the complexity of the task. For the SINGLE reaction time tests, the variation of the estimated delays is smaller than for the INDIVIDUAL test. The deviation for the delays estimated from the BLANK-OUT test is even larger, which indicates that virtual stick balancing might be a more complex task than the two-choice individual reaction time tests.

## 4.2. Critical parameter of the human controller

Blue markers in figure 6 show the experimentally determined critical lengths as a function of the overall reaction delay $\tau = \tau_N + \tau_M + \tau_{added}$ for all the 27 subjects. The neural reaction delay $\tau_N$ of the subjects was chosen to be the average of the results during the blank-out tests. The standard deviation of the estimated neural delay is indicated by an error bar. The uncertainty in the critical length is equal to $\Delta L_5 = 0.0625$ m in each case. The red line indicates the parabola $a_1\tau^2$ fitted on the measured data. This curve can be associated with the critical length $L_{crit}|_{\delta>0}$ corresponding to non-zero stability radii. The goodness of fit for the individual subjects is evaluated using the coefficient of determination $R^2$. The value of $R^2$ ranges from 0 to 1. $R^2 = 1$ means that the data points exactly fit on a parabola of the form $a_1\tau^2$. $R^2 = 0$ indicates no second-order relationship between $\tau$ and $L_{crit}$. Subjects in figure 6 are sorted by the

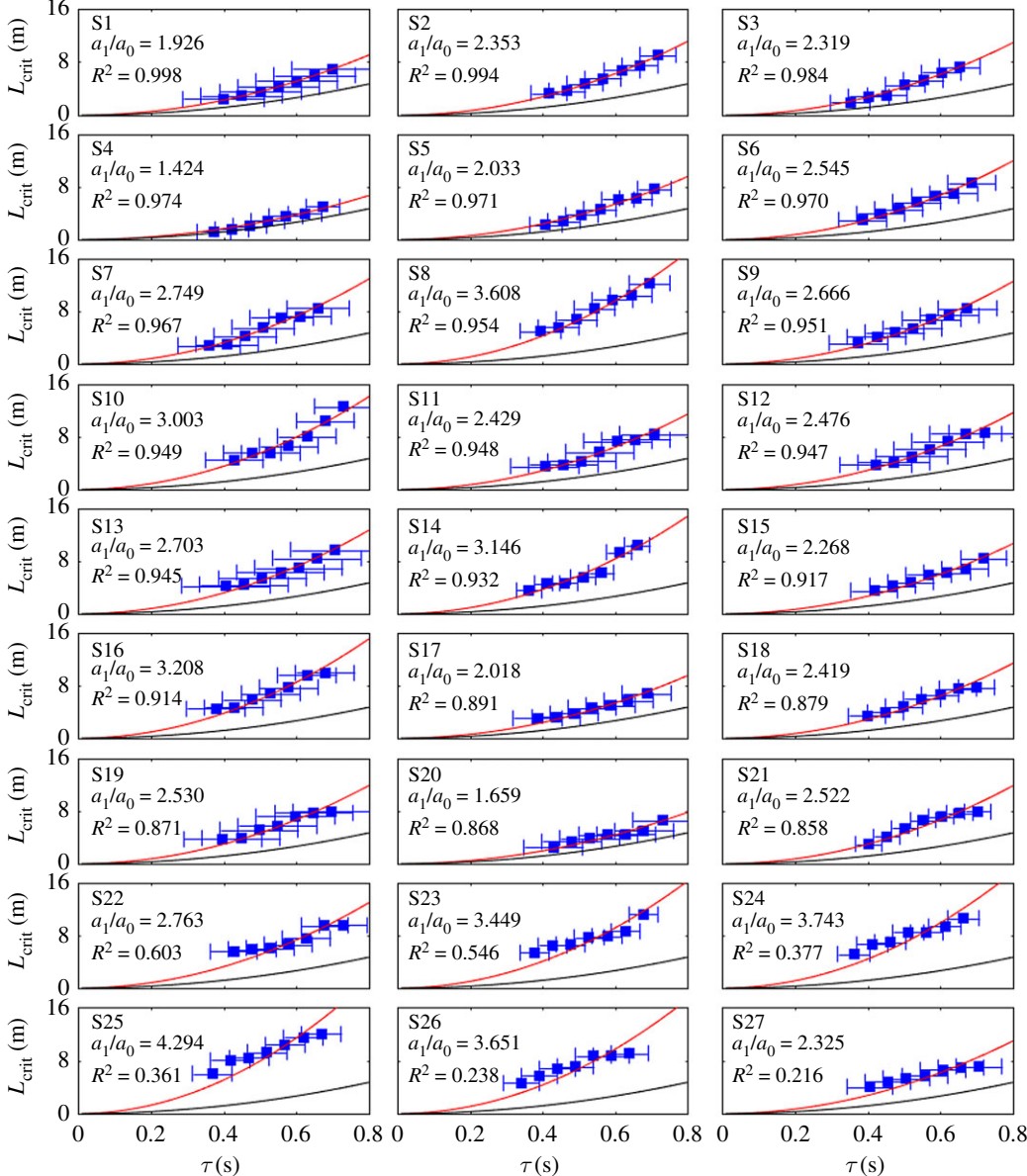

**Figure 6.** Experimentally determined critical lengths as function of the overall reaction time delay. Blue markers and error bars indicate the mean and the standard deviation of the overall delay. Red curve denotes the fitted parabola $L_{\mathrm{crit}}\big|_{\delta>0} = a_1\tau^2$. Black curve denotes the theoretical critical length $L_{\mathrm{crit}}\big|_{\delta=0} = a_0\tau^2$.

corresponding coefficient of determination $R^2$. For 21 out of 27 subjects $R^2 > 0.85$, which supports the assumption of $L_{\mathrm{crit}}$ being proportional to $\tau^2$. The black curve corresponds to the theoretical critical length $L_{\mathrm{crit}}|_{\delta=0} = a_0\tau^2$, where $a_0 = (3/4)g$ (see §2). The difference between the theoretical (black) and the experimental (red) parabolas is characterized by the ratio $a_1/a_0$ (also indicated in each panel in figure 6). Note that the critical length for the subjects is clearly larger than the theoretical critical length $L_{\mathrm{crit}}|_{\delta=0}$, that is, $a_1 > a_0$ for all subjects.

Once the ratio $a_1/a_0$ is determined for the individual subjects, the corresponding stability radii $\delta_{\mathrm{p}}$ and $\delta_{\mathrm{d}}$ can be determined based on figure 2c,d. Note that $\delta_{\mathrm{p}}$ and $\delta_{\mathrm{d}}$ describe the relative width and the relative height of the stable region (figure 1) and the shape of the stable region implies that $\delta_{\mathrm{p}} < \delta_{\mathrm{d}}$ (see §2.3). The ratio $a_1/a_0$ and the corresponding stability radii are shown in figure 7 for all test subjects. The stability radius $\delta_{\mathrm{p}}$ ranges between 3.1 and 27.6% with mean value 14.1%. This gives an upper limit for the combined uncertainty $\varepsilon_{\mathrm{Q,p}}$ associated purely with the perception of the angular position (recall that the system is unstable if $\varepsilon_{\mathrm{Q,p}} > \delta_{\mathrm{p}}$). In other words, in average larger than 14.1% relative error in the control force term $Q_{\mathrm{p}}$ destabilizes the control process. The stability radius $\delta_{\mathrm{d}}$ ranges between 16.8 and 58.2% with mean value 40.3%. This gives an upper limit for the combined uncertainty $\varepsilon_{\mathrm{Q,d}}$ associated

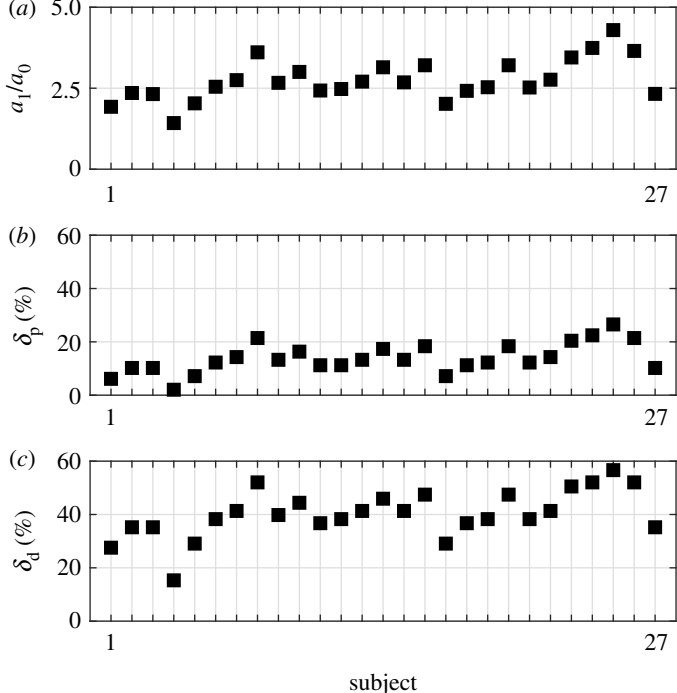

**Figure 7.** Distribution of the coefficient ratio $a_1/a_0$ for the individual test subjects obtained by fitting the measured critical lengths to the parabola $a_1\tau^2$ (a) and the associated stability radii $\delta_\mathrm{p}$ (b) and $\delta_\mathrm{d}$ (c).

purely with the perception of the angular velocity (recall that the system is unstable if $\varepsilon_{\mathrm{Q,d}} > \delta_\mathrm{d}$.) That is, in average larger than 40.3% relative error in $Q_\mathrm{d}$ destabilizes the control process. Thus there is a much larger uncertainty associated with the perception of the angular velocity than the angular displacement. This observation is a straight consequence of the shape of the stable region in figure 1, namely, $\Delta d/d_0 > \Delta p/p_0$. On the neurophysiology side, it is also in agreement with the concept that perception of velocity based on visual feedback is more complex than perception of position [46,47].

## 5. Discussion

The addition of a time delay has been frequently used to identify the nature of the bifurcations that can occur in physiological control mechanisms [48–50]. Here, we have used this approach with a virtual balancing task in order to measure the dependency of $L_\mathrm{crit}$ on $\tau$. We observed that $L_\mathrm{crit} = a_1\tau^2$, where the coefficient $a_1$ is related to the presence of sensorimotor uncertainties. In particular, these uncertainties are related to the size of the stable region in the plane of the control parameters, which corresponds to the stability radii (i.e. allowed static uncertainty of the control parameters). Based on a systematic series of balancing trials by 27 subjects, the overall error associated purely with the angular position was estimated to be around 14%, while the error associated purely with the angular velocity was estimated to be 40%. In the more general case in which sensorimotor uncertainties exist in both angular position and velocity, it is predicted that the uncertainties in angular velocity will be greater.

An obvious source of uncertainty in the perception of the stick position and velocity is originated from the resolution and the refresh rate of the computer screen. The screen resolution was $1920 \times 1080$ and the size of one pixel is $\Delta_\mathrm{pixel} = 0.277$ mm. The refresh rate was $f = 60$ Hz. This results in a resolution of $\Delta\varphi = \Delta_\mathrm{pixel}/L$ in the perception of the angular position and $\Delta\dot\varphi = f\Delta_\mathrm{pixel}/L$ in the angular velocity during a single refresh period. For instance, for a stick of length $L = 5$ m, which appears to be 23 cm long on the screen, $\Delta\varphi = 0.0032°$ and $\Delta\dot\varphi = 0.191° \, \mathrm{s}^{-1}$. The maximum angles during the shortest successfully balanced sticks with zero added delay for all subjects were on average 0.97°, which is significantly larger than the pixel-caused uncertainty. The maximum angular velocities in average were $5.45° \, \mathrm{s}^{-1}$, which is again significantly larger than the pixel-caused uncertainty. These observations support that the resolution and the refresh rate do not significantly contribute to the uncertainty of the overall sensory perception. In other words, the effect of spatial and temporal digitization of the stick is negligible.

Here, we have not considered the effects of stochastic perturbations. Whereas control gains change from trial to trial, the effects of neuronal membrane noise on motor planning are always present. We anticipate that given the large changes in motor planning neural networks, the effects of static changes in the control gains on balancing performance will always be much more significant during the early stages of learning than the effects of noise. This is particular true because of the presence of a central refractory time; that is, changes in the control of a task cannot be made by the nervous system until the previous corrections have been completed [51,52]. Moreover, neuronal membrane noise is likely to be of the same intensity in novice and experts and hence likely has only a small role to play in reducing the variance between repeated trials as skill increases. The advantage of focusing on deterministic uncertainties in the control process is that we can obtain greater insights into the problems faced by a person doing an unpractised task. Nevertheless, although the presented analysis used the concept of static uncertainties according to [18,19,36,37], the results can also be related to stochastic uncertainties (noise) both in neural perception and in motor control. As shown in [53,54], stochastic perturbation has a similar effect on the performance of the control process: the region of parameters for which the system is stable in the presence of noise is typically smaller than the stable region for the nominal (noise-free) system. In this sense, the stability radii can be used to demonstrate the robustness of the system against noise.

Considerations of sensory uncertainties are important for the design of motor tasks for novices, including small children, and for devising safe 'one time' strategies to be used in emergency situations. The goal must be to adjust parameters so that the stability regions in parameter space are larger than the relevant sensory uncertainties. For the special case of stick balancing on the fingertip this can be simply accomplished by making $L$ sufficiently long. However, it is quite possible that the approach we have illustrated here can, with suitable modifications, be applied to other examples of motor and physiological control.

Although the estimated stability radii are related to the specific task of virtual stick balancing, the conclusions might be valid for other tasks that involve the stabilization of an unstable equilibrium. The result that the stability radius for the velocity feedback is larger than that for position feedback is a straight consequence of the ratio of the height and the width of the stable region in figure 1. This emphasizes that the stabilization of an inverted pendulum in the presence of feedback delay is indeed a benchmark problem in human balancing.

Ethics. The research was carried out following the principles of the Declaration of Helsinki. All participants provided informed consent for all research testing.

Data accessibility. The computer code (JAVA) for the virtual stick balancing tests and the data for the stick balancing trials used in this study are available in .mat format at the Dryad Digital Repository at: https://doi.org/10.5061/dryad.41ns1rn9m [45].

Authors' contributions. B.A.K. and T.I. designed and performed the virtual balancing tests. B.A.K. performed the data analysis and the necessary computations. J.M. evaluated the neurophysiological aspects of the results. All authors contributed to the writing of the paper and gave final approval for publication.

Competing interests. We declare we have no competing interest.

Funding. We received no funding for this study.

Acknowledgements. The research reported in this paper was supported by the Higher Education Excellence Program of the Ministry of Human Capacities in the frame of Biotechnology research area of Budapest University of Technology and Economics (BME FIKP-BIO). This research was supported by the Hungarian-French Bilateral Scientific and Technological Cooperation fund under grant no. 2017-2.2.5-TT-FR-2017-00007. J.M. was supported by the William R. Kenan, Jr Charitable trust and a J. T. Oden visiting faculty fellowship while at the Oden Institute for Computational Engineering and Sciences, UT Austin.

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
