## [Reviewer comments · Royal Society Open Science]

Review History

RSOS-191006.R0 (Original submission)

Review form: Reviewer 1

Is the manuscript scientifically sound in its present form?

Yes

Are the interpretations and conclusions justified by the results?

Yes

Is the language acceptable?

Yes

Do you have any ethical concerns with this paper?

No

Have you any concerns about statistical analyses in this paper?

No

Recommendation?

Accept with minor revision (please list in comments)

Comments to the Author(s)

The authors have convincingly addressed the points I had previously raised and have substantially improved the manuscript. I think the manuscript may benefit from clarifying a few remaining points. I have also noted a few typos.

Clarifications

Experimental versus modelling results

The main experimental result is that the critical length scales with τ^2 and that the proportionality constant is 1.4 to 3.6 times larger than would be expected from a PD controller without sensorimotor uncertainties.

It is not an experimental result that velocity uncertainty is larger than position uncertainty for human stick balancing. This is only a modelling result: it is the PD controller itself which is more robust to velocity uncertainty than to position uncertainty. The experimental protocol cannot determine whether the human subjects are more robust to position or velocity uncertainty. This distinction should be made clear both in the abstract and the main text; for example, by saying that, according to the model, the experimental result could be explained either by a 14% position uncertainty or by a 40% velocity uncertainty.

Uncertainties

The title refers to “sensory uncertainties” whereas the abstract refers to “uncertainties during the generation of motor control force”. In the main text, both are used, as well as “static uncertainties related to the system and control parameters” and “uncertainties in the perceived feedback”. It would be useful to use a single terminology, perhaps “sensorimotor uncertainties”? In the main text, it may also be useful to say that you consider static multiplicative error (ex: p.4 l. 27).

Minor points

Notations in the abstract

p.1 l. 30 - 31

It is a bit cumbersome to introduce the notations L_{crit,a_1} directly in the abstract. Please consider rephrasing the 4th and 5th sentence according to: “It is shown that the shortest stick length that human subjects are able to balance is proportional to τ^2 . The proportionality constant can be related to a combined effect of...”

Derivation of $L_{crit} | (\delta > 0) = a_1 \tau^2$

p.5 l.32

It can actually be shown analytically rather than numerically that the critical length associated with a certain δ can be written as $L_{crit} | \delta = a_1 \tau^2$. This can be done by introducing the dimensionless variable $X = \omega \tau$ into equations (3) and (4) to obtain the dimensionless variables $P(X)$ and $D(X)$:

$$P(X) = \tau^2 p(\omega \tau) = (\tau^2 a + X^2) \cos(X)$$

$$D(X) = \tau d(\omega \tau) = (\tau^2 a + X^2) / X \sin(X)$$

A certain δ then corresponds to a unique value of $\tau^2 a = f(\delta)$, with $f(0) = 2$. This explains firstly that $L_{crit} | (\delta = 0) = 3/4 g \tau^2$ and secondly that $L_{crit} | (\delta > 0) = 3/(2f(\delta)) g \tau^2$.

Time delay during blank out tests

p. 9 l. 29 - 46

In their response, the authors indicate that there is no significant difference between calculating the time delay during blank out tests using 1) the indicator function for each trial or 2) the average corrective acceleration across trials of a given subject. The paper could gain in clarity and conciseness by replacing method 1) with method 2).

Typos

1. Introduction

p.1 l. 45: Please replace “novel voluntary motor task” by either “a novel voluntary motor task” or “novel voluntary motor tasks”.

p.2 l. 38: Please replace “increases” by “increase”.

2.1 Stability

p.3 l. 40: Please replace “root” by “roots”.

2.3 Robustness against static uncertainties

p.4 l.53: Please replace “in a combination” by “in combination”.

4.2 Critical parameter of the human controller

p.11 l. 60: Please replace “as function” by “as a function”.

p.12 l. 8: Please replace “shown by errorbar” by “indicated by an errobar”.

p.12 l. 9: Please replace “Red line” by “The red line”.

p.12 l. 11: Please replace “subjected” by “corresponding”.

p.12 l. 18: Please replace “Black curve” by “The black curve”.

p.12 l. 27: Please replace “describes” by “describe”.

5. Discussion

p.14 l. 49 and l. 51: Please replace “in average” by “on average”.

p.14 l. 52: It seems some text is missing in “This supports [missing text] that”.

p.14 l. 59: Please replace “change trial to trial” by “change from trial to trial”.

p. 15 l. 22: Please replace “In these sense” by “In this sense”.

Review form: Reviewer 2

Is the manuscript scientifically sound in its present form?

Yes

Are the interpretations and conclusions justified by the results?

No

Is the language acceptable?

Yes

Do you have any ethical concerns with this paper?

No

Have you any concerns about statistical analyses in this paper?

No

Recommendation?

Major revision is needed (please make suggestions in comments)

Comments to the Author(s)

The study starts from the mathematical model of a pendulum-balancing task using a PD controller with a delayed feedback. The balancing task was performed by 27 human subjects and the experimental results showed that the “experimental” critical length for the given time delay was higher than the theoretical one. The authors explain this by the existence of a static uncertainty related to the estimation of angular position and velocity (sensory uncertainty) as well as the realization of the related control inputs (motor uncertainty).

This is an interesting study. However, there are several important drawbacks that are explained below.

The section 2.3. Robustness against static uncertainties is confusing, since the authors use the terms inconsistently. For example, the critical length is defined in the previous section as the length where “the stable region disappears”, therefore the size of the stable region for L_{crit} is zero. However, in the section 2.3, the authors use the terms such as “the critical length for a given stability radius”, which is not necessarily zero. I suggest that the authors make a distinction between “theoretical” L_{crit} , which is defined based on the zero stability region, and “experimental” L_{crit} , defined as the length for which a human subject loses the control. The latter could be also termed L_{crit} “in the presence of static uncertainties”.

Along the same lines, in this section, the authors refer interchangeably to the size of the stability region and to the uncertainty radii (region), and they even use the same symbols to refer to both. I recommend clearly separating these notions. The size of the stability region is determined by the parameters τ and L that change during the experiment, as shown in Fig. 1 and expressions 3 and 4. The uncertainty radii is, however, determined by the magnitude of the static uncertainty that is supposedly constant. When the size of the stability region, which for the given τ decreases for decreasing L , drops below the region of uncertainty, the system cannot be stabilized (despite the fact that “theoretical” stability region is not zero). All in all, separate and clearly define those terms. I recommend illustrating both regions in the figure. Finally, since these are crucial terms in the study, please give analytic expressions that relate “experimental” L_{crit} and the size of uncertainty, as well as, size of uncertainty and the ratio a_1/a_0 (i.e., essentially, the expressions that are used to generate Fig. 2).

Why not using SINGLE or INDIVIDUAL to estimate the neural delay instead of BLANK OUT? The latter is related to several problems that the authors already acknowledge. In general, in this test, it is likely that the reaction time is affected by the use of the task dynamics. Namely, the state of the pendulum before the blank out likely affected the reaction time when the feedback has been restored (e.g., if the stick was fast falling, the subject knew that they needed to prepare a strong corrective action).

The authors state that “The mean reaction times in the BLANK OUT test (270 ms in average) corresponds to that of the INDIVIDUAL tests.” However, I suggest that these two results are compared statistically to test if there are statistically significant differences between the two reaction times.

Finally, it is not clear to me how to interpret the main outcome of the study “the average uncertainty in the control force associated with the angular position is 14% while in that of the angular velocity is 40%”. What does this tell us? The indicated uncertainties are a mixture of sensory and motor errors, and it seems that these two components cannot be discriminated. In addition, by definition in the section 2.3., those are static errors. Why would a human subject make systematic errors of that size? From the theoretical model as well as from Fig. 2, the uncertainty radii for position and velocity are coupled (a single ratio a_1/a_0 gives both values).

However, the control could have failed (loss of stability) either due to uncertainty in position or in velocity or indeed in both. This cannot be deduced by the model.

Decision letter (RSOS-191006.R0)

30-Jul-2019

Dear Dr Insperger,

The editors assigned to your paper ("Virtual stick balancing: Sensory uncertainties in angular displacement and velocity") have now received comments from reviewers. We would like you to revise your paper in accordance with the referee and Associate Editor suggestions which can be found below (not including confidential reports to the Editor). Please note this decision does not guarantee eventual acceptance.

Please submit a copy of your revised paper before 22-Aug-2019. Please note that the revision deadline will expire at 00.00am on this date. If we do not hear from you within this time then it will be assumed that the paper has been withdrawn. In exceptional circumstances, extensions may be possible if agreed with the Editorial Office in advance. We do not allow multiple rounds of revision so we urge you to make every effort to fully address all of the comments at this stage. If deemed necessary by the Editors, your manuscript will be sent back to one or more of the original reviewers for assessment. If the original reviewers are not available, we may invite new reviewers.

- Data accessibility

It is a condition of publication that all supporting data are made available either as supplementary information or preferably in a suitable permanent repository. The data accessibility section should state where the article's supporting data can be accessed. This section should also include details, where possible of where to access other relevant research materials

such as statistical tools, protocols, software etc can be accessed. If the data have been deposited in an external repository this section should list the database, accession number and link to the DOI for all data from the article that have been made publicly available. Data sets that have been deposited in an external repository and have a DOI should also be appropriately cited in the manuscript and included in the reference list.

If you wish to submit your supporting data or code to Dryad (<http://datadryad.org/>), or modify your current submission to dryad, please use the following link:
<http://datadryad.org/submit?journalID=RSOS&manu=RSOS-191006>

- **Competing interests**

- **Authors' contributions**

- **Acknowledgements**

- **Funding statement**

Kind regards,

Alice Power

Editorial Coordinator

on behalf of Dr Manoj Srinivasan (Associate Editor) and R. Kerry Rowe (Subject Editor)

Associate Editor's comments (Dr Manoj Srinivasan):

The reviewers see merit in the article and have provided some further questions and suggestions for improvement, mostly minor. We invite a revised version addressing these concerns.

Comments to Author:

Reviewers' Comments to Author:

Reviewer: 1

Comments to the Author(s)

The authors have convincingly addressed the points I had previously raised and have substantially improved the manuscript. I think the manuscript may benefit from clarifying a few remaining points. I have also noted a few typos.

Clarifications

Experimental versus modelling results

The main experimental result is that the critical length scales with τ^2 and that the proportionality constant is 1.4 to 3.6 times larger than would be expected from a PD controller without sensorimotor uncertainties.

It is not an experimental result that velocity uncertainty is larger than position uncertainty for human stick balancing. This is only a modelling result: it is the PD controller itself which is more robust to velocity uncertainty than to position uncertainty. The experimental protocol cannot determine whether the human subjects are more robust to position or velocity uncertainty. This distinction should be made clear both in the abstract and the main text; for example, by saying that, according to the model, the experimental result could be explained either by a 14% position uncertainty or by a 40% velocity uncertainty.

Uncertainties

The title refers to "sensory uncertainties" whereas the abstract refers to "uncertainties during the generation of motor control force". In the main text, both are used, as well as "static uncertainties related to the system and control parameters" and "uncertainties in the perceived feedback". It would be useful to use a single terminology, perhaps "sensorimotor uncertainties"?

In the main text, it may also be useful to say that you consider static multiplicative error (ex: p.4 l. 27).

Minor points

Notations in the abstract

p.1 l. 30 - 31

It is a bit cumbersome to introduce the notations L_{crit,a_1} directly in the abstract. Please consider rephrasing the 4th and 5th sentence according to: "It is shown that the shortest stick length that human subjects are able to balance is proportional to τ^2 . The proportionality constant can be related to a combined effect of..."

Derivation of $L_{crit} | (\delta > 0) = a_1 \tau^2$

p.5 l.32

It can actually be shown analytically rather than numerically that the critical length associated with a certain δ can be written as $L_{crit} | \delta = a_1 \tau^2$. This can be done by introducing the dimensionless variable $X = \omega \tau$ into equations (3) and (4) to obtain the dimensionless variables $P(X)$ and $D(X)$:

$$P(X) = \tau^2 p(\omega \tau) = (\tau^2 a + X^2) \cos\left(\frac{\omega \tau}{X}\right)$$

$$D(X) = \tau d(\omega \tau) = (\tau^2 a + X^2)/X \sin\left(\frac{\omega \tau}{X}\right)$$

A certain δ then corresponds to a unique value of $\tau^2 a = f(\delta)$, with $f(0) = 2$. This explains firstly that $L_{crit} | (\delta = 0) = 3/4 g \tau^2$ and secondly that $L_{crit} | (\delta > 0) = 3/(2f(\delta)) g \tau^2$.

Time delay during blank out tests

p. 9 l. 29 - 46

In their response, the authors indicate that there is no significant difference between calculating the time delay during blank out tests using 1) the indicator function for each trial or 2) the average corrective acceleration across trials of a given subject. The paper could gain in clarity and conciseness by replacing method 1) with method 2).

Typos

1. Introduction

p.1 l. 45: Please replace "novel voluntary motor task" by either "a novel voluntary motor task" or "novel voluntary motor tasks".

p.2 l. 38: Please replace "increases" by "increase".

2.1 Stability

p.3 l. 40: Please replace "root" by "roots".

2.3 Robustness against static uncertainties

p.4 l.53: Please replace "in a combination" by "in combination".

4.2 Critical parameter of the human controller

p.11 l. 60: Please replace "as function" by "as a function".

p.12 l. 8: Please replace "shown by errorbar" by "indicated by an errobar".

p.12 l. 9: Please replace "Red line" by "The red line".

p.12 l. 11: Please replace "subjected" by "corresponding".

p.12 l. 18: Please replace "Black curve" by "The black curve".

p.12 l. 27: Please replace "describes" by "describe".

5. Discussion

p.14 l. 49 and l. 51: Please replace "in average" by "on average".

p.14 l. 52: It seems some text is missing in "This supports [missing text] that".

p.14 l. 59: Please replace "change trial to trial" by "change from trial to trial".

p. 15 l. 22: Please replace "In these sense" by "In this sense".

Reviewer: 2

Comments to the Author(s)

The study starts from the mathematical model of a pendulum-balancing task using a PD controller with a delayed feedback. The balancing task was performed by 27 human subjects and the experimental results showed that the "experimental" critical length for the given time delay was higher than the theoretical one. The authors explain this by the existence of a static uncertainty related to the estimation of angular position and velocity (sensory uncertainty) as well as the realization of the related control inputs (motor uncertainty).

This is an interesting study. However, there are several important drawbacks that are explained below.

The section 2.3. Robustness against static uncertainties is confusing, since the authors use the terms inconsistently. For example, the critical length is defined in the previous section as the length where “the stable region disappears”, therefore the size of the stable region for L_{crit} is zero. However, in the section 2.3, the authors use the terms such as “the critical length for a given stability radius”, which is not necessarily zero. I suggest that the authors make a distinction between “theoretical” L_{crit} , which is defined based on the zero stability region, and “experimental” L_{crit} , defined as the length for which a human subject loses the control. The latter could be also termed L_{crit} “in the presence of static uncertainties”.

Along the same lines, in this section, the authors refer interchangeably to the size of the stability region and to the uncertainty radii (region), and they even use the same symbols to refer to both. I recommend clearly separating these notions. The size of the stability region is determined by the parameters τ and L that change during the experiment, as shown in Fig. 1 and expressions 3 and 4. The uncertainty radii is, however, determined by the magnitude of the static uncertainty that is supposedly constant. When the size of the stability region, which for the given τ decreases for decreasing L , drops below the region of uncertainty, the system cannot be stabilized (despite the fact that “theoretical” stability region is not zero). All in all, separate and clearly define those terms. I recommend illustrating both regions in the figure. Finally, since these are crucial terms in the study, please give analytic expressions that relate “experimental” L_{crit} and the size of uncertainty, as well as, size of uncertainty and the ratio a_1/a_0 (i.e., essentially, the expressions that are used to generate Fig. 2).

Why not using SINGLE or INDIVIDUAL to estimate the neural delay instead of BLANK OUT? The latter is related to several problems that the authors already acknowledge. In general, in this test, it is likely that the reaction time is affected by the use of the task dynamics. Namely, the state of the pendulum before the blank out likely affected the reaction time when the feedback has been restored (e.g., if the stick was fast falling, the subject knew that they needed to prepare a strong corrective action).

The authors state that “The mean reaction times in the BLANK OUT test (270 ms in average) corresponds to that of the INDIVIDUAL tests.” However, I suggest that these two results are compared statistically to test if there are statistically significant differences between the two reaction times.

Finally, it is not clear to me how to interpret the main outcome of the study “the average uncertainty in the control force associated with the angular position is 14% while in that of the angular velocity is 40%”. What does this tell us? The indicated uncertainties are a mixture of sensory and motor errors, and it seems that these two components cannot be discriminated. In addition, by definition in the section 2.3., those are static errors. Why would a human subject make systematic errors of that size? From the theoretical model as well as from Fig. 2, the uncertainty radii for position and velocity are coupled (a single ratio a_1/a_0 gives both values). However, the control could have failed (loss of stability) either due to uncertainty in position or in velocity or indeed in both. This cannot be deduced by the model.

Author's Response to Decision Letter for (RSOS-191006.R0)

See Appendix A.

RSOS-191006.R1 (Revision)

Review form: Reviewer 1

Is the manuscript scientifically sound in its present form?

Yes

Are the interpretations and conclusions justified by the results?

Yes

Is the language acceptable?

Yes

Do you have any ethical concerns with this paper?

No

Have you any concerns about statistical analyses in this paper?

No

Recommendation?

Accept as is

Comments to the Author(s)

This is a very interesting paper which is now fit for publication.

A very minor suggestion: perhaps you could mention that the uncertainty associated with angular position is *at most* 14% and that in angular velocity is *at most* 40% (page 7 lines 42, 43, page 21 lines 16 and 17).

Review form: Reviewer 2

Is the manuscript scientifically sound in its present form?

Yes

Are the interpretations and conclusions justified by the results?

Yes

Is the language acceptable?

Yes

Do you have any ethical concerns with this paper?

No

Have you any concerns about statistical analyses in this paper?

Yes

Recommendation?

Accept with minor revision (please list in comments)

Comments to the Author(s)

The authors have addressed most of my comments in the revision. I have two minor suggestions.

The authors have used three tests to determine the reaction time: INDIVIDUAL, BLANK OUT and SINGLE. In the revised manuscript, they have now compared BLANK OUT and INDIVIDUAL using a t-test on a subject-by-subject basis as it seems. However, it seems to me more logical to compare group means and standard deviations by using ANOVA with a post hoc test for the pairwise comparisons (or their non-parametric equivalents). This is a usual approach when you have multiple groups.

I would also propose that a similar statistical comparison is performed between Δ_d and Δ_p to confirm that these values are indeed statistically significantly different (since this is the most important conclusion of the study).

I still struggle a bit to grasp the potential relevance and future application of the main conclusion of the study, outside of the specific experiment presented in the current manuscript. Is the fact that the stability radii for velocity are higher than for position universal characteristics of the human motor control or this is a result that is after all specific to the pendulum balancing task? Are the estimates of those radii 14% and 40% universally valid or again specific to this test? If these numbers are just an illustration, is the main contribution of the present paper in the methodology (and not the actual values)? The authors discuss a bit potential future application but this is rather general. I would suggest the authors to expand on the relevance of the present study and their conclusions.

Decision letter (RSOS-191006.R1)

22-Oct-2019

Dear Dr Insperger,

On behalf of the Editors, I am pleased to inform you that your Manuscript RSOS-191006.R1 entitled "Virtual stick balancing: Sensorimotor uncertainties related to angular displacement and velocity" has been accepted for publication in Royal Society Open Science subject to minor revision in accordance with the referee suggestions. Please find the referees' comments at the end of this email.

The reviewers and Subject Editor have recommended publication, but also suggest some minor revisions to your manuscript. Therefore, I invite you to respond to the comments and revise your manuscript.

- Ethics statement

- Data accessibility

It is a condition of publication that all supporting data are made available either as supplementary information or preferably in a suitable permanent repository. The data accessibility section should state where the article's supporting data can be accessed. This section

should also include details, where possible of where to access other relevant research materials such as statistical tools, protocols, software etc can be accessed. If the data has been deposited in an external repository this section should list the database, accession number and link to the DOI for all data from the article that has been made publicly available. Data sets that have been deposited in an external repository and have a DOI should also be appropriately cited in the manuscript and included in the reference list.

If you wish to submit your supporting data or code to Dryad (<http://datadryad.org/>), or modify your current submission to dryad, please use the following link:
<http://datadryad.org/submit?journalID=RSOS&manu=RSOS-191006.R1>

- **Competing interests**

- **Authors' contributions**

- **Acknowledgements**

- **Funding statement**

Because the schedule for publication is very tight, it is a condition of publication that you submit the revised version of your manuscript before 31-Oct-2019. Please note that the revision deadline will expire at 00.00am on this date. If you do not think you will be able to meet this date please let me know immediately.

To revise your manuscript, log into <https://mc.manuscriptcentral.com/rsos> and enter your Author Centre, where you will find your manuscript title listed under "Manuscripts with Decisions". Under "Actions," click on "Create a Revision." You will be unable to make your

revisions on the originally submitted version of the manuscript. Instead, revise your manuscript and upload a new version through your Author Centre.

Kind regards,
Lianne Parkhouse
Editorial Coordinator
Royal Society Open Science
openscience@royalsociety.org

on behalf of Dr Manoj Srinivasan (Associate Editor) and R. Kerry Rowe (Subject Editor)
openscience@royalsociety.org

Associate Editor Comments to Author (Dr Manoj Srinivasan):

Please consider revising the article to address the minor comments of one of the reviewers in your final version.

Reviewer comments to Author:

Reviewer: 1

Comments to the Author(s)

This is a very interesting paper which is now fit for publication.

A very minor suggestion: perhaps you could mention that the uncertainty associated with angular position is *at most* 14% and that in angular velocity is *at most* 40% (page 7 lines 42, 43, page 21 lines 16 and 17).

Reviewer: 2

Comments to the Author(s)

The authors have addressed most of my comments in the revision. I have two minor suggestions.

The authors have used three tests to determine the reaction time: INDIVIDUAL, BLANK OUT and SINGLE. In the revised manuscript, they have now compared BLANK OUT and INDIVIDUAL using a t-test on a subject-by-subject basis as it seems. However, it seems to me more logical to compare group means and standard deviations by using ANOVA with a post hoc test for the pairwise comparisons (or their non-parametric equivalents). This is a usual approach when you have multiple groups.

I would also propose that a similar statistical comparison is performed between δ_d and δ_p to confirm that these values are indeed statistically significantly different (since this is the most important conclusion of the study).

I still struggle a bit to grasp the potential relevance and future application of the main conclusion of the study, outside of the specific experiment presented in the current manuscript. Is the fact that the stability radii for velocity are higher than for position universal characteristics of the human motor control or this is a result that is after all specific to the pendulum balancing task? Are the estimates of those radii 14% and 40% universally valid or again specific to this test? If these numbers are just an illustration, is the main contribution of the present paper in the methodology (and not the actual values)? The authors discuss a bit potential future application but this is rather general. I would suggest the authors to expand on the relevance of the present study and their conclusions.

Author's Response to Decision Letter for (RSOS-191006.R1)

See Appendix B.

Decision letter (RSOS-191006.R2)

01-Nov-2019

Dear Dr Insperger,

I am pleased to inform you that your manuscript entitled "Virtual stick balancing: Sensorimotor uncertainties related to angular displacement and velocity" is now accepted for publication in Royal Society Open Science.

Kind regards,
Lianne Parkhouse
Editorial Coordinator
Royal Society Open Science
openscience@royalsociety.org

on behalf of Dr Manoj Srinivasan (Associate Editor) and R. Kerry Rowe (Subject Editor)
openscience@royalsociety.org

Appendix A

Response to reviewers

Reviewer: 1

Comments to the Author(s)

The authors have convincingly addressed the points I had previously raised and have substantially improved the manuscript. I think the manuscript may benefit from clarifying a few remaining points. I have also noted a few typos.

Authors:

We thank the reviewer for the useful suggestions. We reply to the issues raised by the reviewer below. We have highlighted the corresponding modifications in the manuscript by red color.

Clarifications

Experimental versus modelling results

The main experimental result is that the critical length scales with τ^2 and that the proportionality constant is 1.4 to 3.6 times larger than would be expected from a PD controller without sensorimotor uncertainties.

It is not an experimental result that velocity uncertainty is larger than position uncertainty for human stick balancing. This is only a modelling result: it is the PD controller itself which is more robust to velocity uncertainty than to position uncertainty. The experimental protocol cannot determine whether the human subjects are more robust to position or velocity uncertainty.

This distinction should be made clear both in the abstract and the main text; for example, by saying that, according to the model, the experimental result could be explained either by a 14% position uncertainty or by a 40% velocity uncertainty.

Authors:

The reviewer is right, this point was not clarified enough. The robustness against uncertainty in the velocity and the position originates from the delayed PD model, i.e., from the shape of the stable region. In other words, the delayed PD model allows larger uncertainty in the velocity than in the position, while on the other side, it is also in agreement with the concept that perception of velocity based on visual feedback is more complex than perception of position. Nevertheless, the separation of the effects of the uncertainty in the position and in the velocity was done for mathematical convenience. We have modified the abstract and the text throughout the paper in order to highlight the above issues.

Uncertainties

The title refers to “sensory uncertainties” whereas the abstract refers to “uncertainties during the generation of motor control force”. In the main text, both are used, as well as “static uncertainties related to the system and control parameters” and “uncertainties in the perceived feedback”. It would be useful to use a single terminology, perhaps “sensorimotor uncertainties”?

In the main text, it may also be useful to say that you consider static multiplicative error (ex: p.4 l. 27).

Authors:

Thank you for this comment. We have changed the wording to “sensorimotor uncertainties” as suggested by the reviewer.

We have highlighted after eq. (9) that the static error in the realization of the control force can be considered as a multiplicative error in the actual control force.

Minor points

Notations in the abstract p.1 l. 30 – 31

It is a bit cumbersome to introduce the notations L_{crit,a_1} directly in the abstract. Please consider rephrasing the 4th and 5th sentence according to : “It is shown that the shortest stick length that human subjects are able to balance is proportional to τ^2 . The proportionality constant can be related to a combined effect of...”

Authors: Thank you, we have modified the abstract as suggested and removed the notations for the parameters.

Derivation of $L_{crit} |(\delta > 0) = a_1 \tau^2$

p.5 l.32

It can actually be shown analytically rather than numerically that the critical length associated with a certain δ can be written as $L_{crit} | \delta = a_1 \tau^2$. This can be done by introducing the dimensionless variable $X = \omega\tau$ into equations (3) and (4) to obtain the dimensionless variables $P(X)$ and $D(X)$:

$$P(X) = \tau^2 p(\omega\tau) = (\tau^2 a + X^2) \cos^2(X)$$

$$D(X) = \tau d(\omega\tau) = (\tau^2 a + X^2)/X \sin^2(X)$$

A certain δ then corresponds to a unique value of $\tau^2 a = f(\delta)$, with $f(0) = 2$. This explains firstly that $L_{crit} |(\delta = 0) = 3/4 g\tau^2$ and secondly that $L_{crit} |(\delta > 0) = 3/(2f(\delta)) g\tau^2$.

Authors:

Indeed, $a_1 = 3/(2f(\delta)) g$, where $f(\delta)$ can be given by solving the system of nonlinear equation numerically. This is why we first mentioned that it is a numerical solution. In the revised version we follow the reviewer’s suggestion and have modified the text accordingly in page 5.

Time delay during blank out tests

p. 9 l. 29 - 46

In their response, the authors indicate that there is no significant difference between calculating the time delay during blank out tests using 1) the indicator function for each trial or 2) the average corrective acceleration across trials of a given subject. The paper could gain in clarity and conciseness by replacing method 1) with method 2).

Authors:

Actually, the time delay was estimated using the indicator function in order to get an objective estimation for all the different blankout trials, since determining the delay using the average corrective acceleration was found to be subjective. We have added these comments before eq. (17).

Typos

1. Introduction

p.1 l. 45: Please replace “novel voluntary motor task” by either “a novel voluntary motor task” or “novel voluntary motor tasks”.

p.2 l. 38: Please replace “increases” by “increase”.

2.1 Stability

p.3 l. 40: Please replace “root” by “roots”.

2.3 Robustness against static uncertainties

p.4 l.53: Please replace “in a combination” by “in combination”.

4.2 Critical parameter of the human controller

p.11 l. 60: Please replace “as function” by “as a function”.

p.12 l. 8: Please replace “shown by errorbar” by “indicated by an errorbar”.

p.12 l. 9: Please replace “Red line” by “The red line”.

p.12 l. 11: Please replace “subjected” by “corresponding”.
p.12 l. 18: Please replace “Black curve” by “The black curve”.
p.12 l. 27: Please replace “describes” by “describe”.

5. Discussion

p.14 l. 49 and l. 51: Please replace “in average” by “on average”.
p.14 l. 52: It seems some text is missing in “This supports [missing text] that”.
p.14 l. 59: Please replace “change trial to trial” by “change from trial to trial”.
p. 15 l. 22: Please replace “In these sense” by “In this sense”.

Authors:

Thank you for finding all these typos, we have corrected all of them (these were not highlighted by red in the revised version).

Reviewer: 2

Comments to the Author(s)

The study starts from the mathematical model of a pendulum-balancing task using a PD controller with a delayed feedback. The balancing task was performed by 27 human subjects and the experimental results showed that the “experimental” critical length for the given time delay was higher than the theoretical one. The authors explain this by the existence of a static uncertainty related to the estimation of angular position and velocity (sensory uncertainty) as well as the realization of the related control inputs (motor uncertainty).

This is an interesting study. However, there are several important drawbacks that are explained below.

Authors:

We thank the reviewer for the useful suggestions. We reply to the issues raised by the reviewer below. We have highlighted the corresponding modifications in the manuscript by red color.

The section 2.3. Robustness against static uncertainties is confusing, since the authors use the terms inconsistently. For example, the critical length is defined in the previous section as the length where “the stable region disappears”, therefore the size of the stable region for L_{crit} is zero. However, in the section 2.3, the authors use the terms such as “the critical length for a given stability radius”, which is not necessarily zero. I suggest that the authors make a distinction between “theoretical” L_{crit} , which is defined based on the zero stability region, and “experimental” L_{crit} , defined as the length for which a human subject loses the control. The latter could be also termed L_{crit} “in the presence of static uncertainties”.

Authors:

We have changed the title of sections 2.2 and 2.3 and added some extra explanation on the theoretical critical length without uncertainty and on the critical length in the presence of uncertainties. The former one is indicated by $L_{crit|\delta=0}$ while the latter is indicated by $L_{crit|\delta>0}$ throughout the paper. This is also mentioned in section 2.3 now.

We also introduced a new paragraph in the introduction on application of stability radius in robust stability analysis in the presence of static uncertainties.

Along the same lines, in this section, the authors refer interchangeably to the size of the stability region and to the uncertainty radii (region), and they even use the same

symbols to refer to both. I recommend clearly separating these notions. The size of the stability region is determined by the parameters τ and L that change during the experiment, as shown in Fig. 1 and expressions 3 and 4. The uncertainty radii is, however, determined by the magnitude of the static uncertainty that is supposedly constant. When the size of the stability region, which for the given τ decreases for decreasing L , drops below the region of uncertainty, the system cannot be stabilized (despite the fact that “theoretical” stability region is not zero). All in all, separate and clearly define those terms. I recommend illustrating both regions in the figure. Finally, since these are crucial terms in the study, please give analytic expressions that relate “experimental” L_{crit} and the size of uncertainty, as well as, size of uncertainty and the ratio a_1/a_0 (i.e., essentially, the expressions that are used to generate Fig. 2).

Authors:

Uncertainty radii and stability radii were indeed mixed in Figure 2. We have corrected the terms and modified the manuscript to make the difference clear. The size of the stability region is indicated by Δp and Δd , while the associated stability radii by δ_p and δ_d . In order to make the difference more clear, we have modified Figure 1 and added the robust stability curves on panel (b). The uncertainty radii (or ratio) is indicated by ε_p and ε_d . and it is clearly distinguished from δ_p and δ_d . We have added an extra comment on this after eq.(10) and modified the caption of Figure 2.

We have added comments on the derivation of the formulas plotted in Figure 2. However, we did not added a detailed derivation, since the coefficient a_1 can be determined numerically by solving a system of nonlinear equations. This point was also raised by Reviewer 1.

Why not using SINGLE or INDIVIDUAL to estimate the neural delay instead of BLANK OUT? The latter is related to several problems that the authors already acknowledge. In general, in this test, it is likely that the reaction time is affected by the use of the task dynamics. Namely, the state of the pendulum before the blank out likely affected the reaction time when the feedback has been restored (e.g., if the stick was fast falling, the subject knew that they needed to prepare a strong corrective action).

Authors:

Regarding the identification of the delay, we have used the delays obtained from the BLANKOUT tests because it was determined during performing the same task. The SINGLE and INDIVIDUAL tests were found to be less complex. We agree that there are many uncertainties in determining the delay from the BLANKOUT tests, therefore we applied an objective method by using the indicator function in order to get an objective estimation for all the different blankout trials. We have added these comments before eq. (17). SINGLE and INDIVIDUAL tests were presented only for validation.

The authors state that “The mean reaction times in the BLANK OUT test (270 ms in average) corresponds to that of the INDIVIDUAL tests.” However, I suggest that these two results are compared statistically to test if there are statistically significant differences between the two reaction times.

Authors:

We have added the details of a statistical comparison in section 4.1 where we discuss the result of the different reaction time tests.

Finally, it is not clear to me how to interpret the main outcome of the study “the average uncertainty in the control force associated with the angular position is 14% while in that of the angular velocity is 40%”. What does this tell us? The indicated uncertainties are a mixture of sensory and motor errors, and it seems that these two components

cannot be discriminated. In addition, by definition in the section 2.3., those are static errors. Why would a human subject make systematic errors of that size? From the theoretical model as well as from Fig. 2, the uncertainty radii for position and velocity are coupled (a single ratio a_1/a_0 gives both values). However, the control could have failed (loss of stability) either due to uncertainty in position or in velocity or indeed in both. This cannot be deduced by the model.

Authors:

Indeed, the uncertainties associated with position and angular velocity are not independent in the model and a single a_1/a_0 ratio gives both values, namely the value of $a_1/a_0 \approx 2.5$ gives $\delta_p \approx 14\%$ and $\delta_d \approx 40\%$. This means that the system becomes unstable either for 14% error in the position or for 40% error in the velocity.

Indeed, a time-dependent or stochastic uncertainty may be more realistic, than a static error model. However, attention to static uncertainties is also important since these uncertainties determine the types of feedback control mechanisms that will be effective for control in uncertain environments. We have added several comments in the Introduction to point out the importance of determining the static uncertainties. One reason for the static model is that it can be analyzed easily using the concept of stability radii. Another reason is that stochastic perturbation has a similar effect on the performance of the control process: the region of parameters for which the system is stable in the presence of noise is typically smaller than the stable region for the nominal (noise-free) system. In this sense, the stability radii can be used to demonstrate the robustness of the system against noise.

We have added comments on the above issues in section 2.3 and also in the Discussion.

Appendix B

Dear Professor R. Kerry Rowe,
Dear professor Manoj Srinivasan,

We are submitting our revised manuscript ID RSOS-191006.R1, which was submitted for publication as research paper to the Royal Society Open Science.

We have addressed the comments of the reviewers in separate files. All the changes in the manuscript have been highlighted by blue color.

Sincerely,
Tamas Inspurger
corresponding author

Response to reviewers

Reviewer: 1 Comments to the Author(s)

This is a very interesting paper which is now fit for publication.
A very minor suggestion: perhaps you could mention that the uncertainty associated with angular position is *at most* 14% and that in angular velocity is *at most* 40% (page 7 lines 42, 43, page 21 lines 16 and 17).

Authors:

We thank the reviewer for this comment, we have modified the sentence as suggested.

Reviewer: 2 Comments to the Author(s)

The authors have addressed most of my comments in the revision. I have two minor suggestions.

The authors have used three tests to determine the reaction time: INDIVIDUAL, BLANK OUT and SINGLE. In the revised manuscript, they have now compared BLANK OUT and INDIVIDUAL using a t-test on a subject-by-subject basis as it seems. However, it seems to me more logical to compare group means and standard deviations by using ANOVA with a post hoc test for the pairwise comparisons (or their non-parametric equivalents). This is a usual approach when you have multiple groups.

Authors:

We have carried out a repeated measure ANOVA test in order to verify our observation as suggested. The statistical test showed significance between the different tests. (The level of significance was 94% at most). We have added a comment on the result of ANOVA test in section 4.1.

I would also propose that a similar statistical comparison is performed between δ_d and δ_p to confirm that these values are indeed statistically significantly different (since this is the most important conclusion of the study).

Authors:

We do not claim that δ_d and δ_p are statistically significantly different, but their ratio is a consequence of the shape of the stable region. We have emphasized this at several parts of the manuscript. This is also confirmed by t-test, which showed no significant difference between δ_p and δ_d .

I still struggle a bit to grasp the potential relevance and future application of the main conclusion of the study, outside of the specific experiment presented in the current manuscript. Is the fact that the stability radii for velocity are higher than for position universal characteristics of the human motor control or this is a result that is after all specific to the pendulum balancing task? Are the estimates of those radii 14% and 40% universally valid or again specific to this test? If these numbers are just an illustration, is the main contribution of the present paper in the methodology (and not the actual values)? The authors discuss a bit potential future application but this is rather general. I would suggest the authors to expand on the relevance of the present study and their conclusions.

Authors:

The radii 14% for position and 40% for velocity are related to the specific task, and their ratio is originated from the underlying mathematical model, namely, the shape for the stability diagram. Nevertheless, the conclusions might be valid to other tasks that involve the stabilization of an unstable equilibrium, since the governing mathematical model is the same that for the virtual balancing task. The result that the stability radius for the velocity feedback is larger than that for position feedback is a consequence of the ratio of the height and the width of the stable region in Figure. 1. This emphasizes that the stabilization of an inverted pendulum in the presence of feedback delay is indeed a benchmark problem in human balancing.

We have added a comment on this issue in the Discussion.